# A Randomized Controlled Trial of Colistin Combined with Sulbactam: 9 g per Day versus 12 g per Day in the Treatment of Extensively Drug-Resistant *Acinetobacter baumannii* Pneumonia: An Interim Analysis

**DOI:** 10.3390/antibiotics11081112

**Published:** 2022-08-17

**Authors:** Chutchawan Ungthammakhun, Vasin Vasikasin, Dhitiwat Changpradub

**Affiliations:** Division of Infectious Disease, Department of Medicine, Phramongkutklao Hospital, Bangkok 10400, Thailand

**Keywords:** *A. baumannii* XDR pneumonia, mortality rate, colistin, sulbactam

## Abstract

Extensively drug-resistant *A. baumannii* (XDRAB) pneumonia has a high mortality rate in hospitalized patients. One of the recommended treatments is colistin combined with sulbactam; however, the optimal dosage of sulbactam is unclear. In an open-label, superiority, randomized controlled trial, patients diagnosed with XDRAB pneumonia were randomly assigned (1:1) to receive colistin in combination with sulbactam at either 9 g/day or 12 g/day. The primary outcome was the 28-day mortality rate in the intention-to-treat population. A total of 88 patients received colistin in combination with sulbactam at a dosage of either 12 g/day (n = 45) or 9 g/day (n = 43). Trends toward a lower mortality rate were observed in the 12 g/day group at 7 days (11.1% vs. 23.3%), 14 days (33.3% vs. 41.9%), and 28 days (46.7% vs. 58.1%). The microbiological cure rate at day 7 was significantly higher in the 12 g/day group (90.5% vs. 58.1%; *p* = 0.02). Factors associated with mortality at 28 days were asthma, cirrhosis, APACHEII score ≥ 28, and a dosage of sulbactam of 9 g/day for mortality at any timepoint. Treatment with colistin combined with sulbactam at 12 g/day was not superior to the combination treatment with sulbactam at 9 g/day. However, due to being an interim analysis, this trial was underpowered to detect mortality differences.

## 1. Introduction and Study Rationale

*A. baumannii* is an emerging pathogen causing infection in hospitalized patients [1,2,3]. Various diseases of hospitalized patients are caused by this pathogen, such as bloodstream infections, pneumonia, meningitis, urinary tract infections, and surgical site infections [2]. The most common infections are pneumonia (57.6%), followed by blood infections (23.9%) and wound infections (9.1%) [3,4]. Risk factors associated with *A. baumannii* infection are hospitalization at an intensive care unit (ICU), presence of intubation, recent broad-spectrum antibiotic exposure, or surgery [1,2].

Extensively drug-resistant *A. baumannii* (XDRAB) is resistant to multiple antibiotics, including fluoroquinolones, aminoglycosides, carbapenems, and penicillin [5,6]. XDRAB is associated with high mortality rate and length of hospital stay [7]. The mortality rate of patients infected with XDRAB is 26–68% in the presence of infection in any organ [4,8,9] and 20–70% in the presence of pneumonia [10,11,12]. Factors associated with mortality rate are improper or delayed antibiotics administration, an APACHEII score > 15, septic shock, acute renal failure, and age over 58 years [4,13,14,15].

The current treatment of XDRAB is unclear [13]. However, the Infectious Diseases Society of America (IDSA) established guidelines in 2016 for the treatment of hospital-acquired pneumonia and ventilator-associated pneumonia [16], recommending carbapenems or sulbactam for the treatment of *A. baumannii* pneumonia if susceptible. Unfortunately, most XDRAB isolates are becoming resistant to these antibiotics and are usually only susceptible to polymyxin antibiotics, such as polymyxin B, polymyxin E, and colistin [17]. This is consistent with data from Phramongkutklao Hospital in 2008 [18]. In the case of strains only susceptible to polymyxin, only intravenous polymyxin is recommended by the IDSA [16].

Because of the poor intravenous colistin levels in the lower pulmonary tissue compared to blood levels [19], a beta lactamase inhibitor, sulbactam, has been recently studied. Sulbactam has a unique bactericidal activity, comparable to that of beta-lactam against *A. baumannii*, based in the binding of penicillin binding protein 1 (PBP1) and PBP3 [20]. It has good pharmacokinetics in lung tissue, making it a potential treatment option for *A. baumannii* pneumonia [21]. Treatment for XDRAB patients was modified by prolonging the infusion time, increasing the dosage, or administering drug combinations [14]. Colistin also exerts a synergistic effect with sulbactam or carbapenems on XDRAB [22,23]. Sulbactam in combination to other drugs than colistin such as tigecycline still showed in vitro synergistic effects [24]. Moreover, a combination therapy has a better therapeutic effect than monotherapies [25,26]. The current 2022 IDSA guidance for the treatment of carbapenem-resistant *A. baumannii* recommends a combination therapy with at least two agents with in vitro activity, such as polymyxin, a high dose of ampicillin–sulbactam, or tigecycline, whereas it recommends against the use of polymyxin in combination with meropenem [27].

Many studies have examined the effects of sulbactam in the treatment of patients with XDRAB pneumonia. The dosages of sulbactam ranged from 3 to 12 g/day [11,14,28,29], but the optimal dosage of sulbactam has not been determined [28,30]. The experts recommend that a sulbactam dose of at least 6 g/day is sufficient to treat XDRAB infections [31,32], but some experts suggest a higher dose of sulbactam (9–12 g/day) for the treatment of XDRAB infections characterized by a high MIC of sulbactam [33]. A study of ampicillin/sulbactam at the dosages of 18/9 and 24/12 g/day found no significant complications compared with colistin monotherapy [11,14]. According to in vitro data from Phramongkutklao Hospital to determine the optimal dose of sulbactam in the treatment of critically ill patients with carbapenem-resistant *A. baumannii*, sulbactam at the dose of up to 12 g/day with prolonged infusion may be sufficient for the treatment of *A. baumannii* with high sulbactam MIC [34]. Therefore, a combination therapy of colistin with sulbactam at a dosage of 9 or 12 g/day for treatment of XDRAB pneumonia was administered to determine the optimum dosage of sulbactam in combination with colistin.

## 2. Objectives

The primary objective was to compare the 28-day mortality rate of XDRAB pneumonia patients after treatment with colistin plus sulbactam at a dosage of 9 g/day or 12 g/day at Phramongkutklao Hospital.

The secondary objectives of this study were to compare treatment outcomes in XDRAB pneumonia patients, including 7-day and 14-day mortality rates, microbiological cure rate, length of hospital stay, ICU days, ventilator days, incidence of adverse drug reactions after treatment of colistin and sulbactam at dosages of 9 or 12 g/day, and factors affecting the mortality rate of XDRAB pneumonia patients.

## 3. Materials and Methods

### 3.1. Ethical Considerations

The study protocol followed the guidelines of the Declaration of Helsinki, and ethical approval was obtained from the Institutional Review Board of the Royal Thai Army Medical Department. (Approval number R108h/62).

### 3.2. Study Design and Participants

This study was a single-center, prospective, open-label, randomized controlled trial. Hospitalized patients with pneumonia caused by XDRAB at Phramongkutklao Hospital (1200 beds) were enrolled.

The inclusion criteria were as follows: patients aged ≥18 years who were diagnosed with hospital-acquired pneumonia (HAP) or ventilator-associated pneumonia (VAP) according to the CDC diagnostic criteria, [35] by sputum from a tracheal suction culture, positive for XDRAB and susceptible to colistin. Patients with sputum culture results showing more than one organism, having colistin-resistant XDRAB, unable to receive colistin as empirical therapy or to participate in the study for 14 days, administered colistin before the enrollment for more than 5 days, infection in other organs, preexisting infection before pneumonia diagnosis, pregnant, or a history of colistin or sulbactam within 14 days were excluded from this study. Signed informed consent was obtained from all participants or representative of the patients before enrollment.

This study was planned for a period of two years, from August 2019 to August 2021, but due to the COVID-19 pandemic, the data collected were less than expected.

### 3.3. Randomization and Masking

The participants who were enrolled in this study were randomly assigned to two groups at a 1:1 ratio to receive colistin in combination with sulbactam at either 9 g/day or 12 g/day. A computer-generated randomization was prepared by a statistician. The list was stratified by blocks of four. During the study, information on the drug and dosage was disclosed to the participants, physicians, investigators, and nursing teams.

### 3.4. Procedures

The study medications, colistin at 300 mg loading dose intravenously in combination with sulbactam at a dosage of either 9 g/day or 12 g/day, were given as a 3 g intravenous dosage with a prolonged 3 h infusion every 8 h, or as a 4 g intravenous dosage with a prolonged 3 h infusion every 8 h in 100 mL NSS. Then, the dosages of colistin and sulbactam were adjusted according to creatinine clearance calculated with the Cockcroft–Gault formula [36,37,38]. The patients received the study medication for 7 days, unless they had any contraindication to the study drugs or expired.

Collected information of the study participants included demographic data such as age, gender, BMI and underlying diseases, duration of hospitalization, ventilator days, and ICU days. The disease severity of the patients was based on the APACHE II score [39] and SOFA [40] score at the date of diagnosis and the date of definite treatment. Standard laboratory procedures for sputum cultures were performed as part of clinical routine. Bacterial species were identified by MALDI-TOF (Bruker Daltonics, Bremen, Germany). The method for measuring the MIC values was according to the Clinical and Laboratory Standards Institute (CLSI) [41] guidelines for sulbactam and colistin against XDRAB. Sulbactam MIC were obtained by the Epsilometer test (E-test, Liofilchem, Waltham, MA, United States). The automated broth microdilution test (Sensititre, Thermo Fisher, Cleveland, OH, United States) was used for the MIC breakpoint of other antibiotics including colistin. XDRAB was diagnosed when an isolate resisted to all antibiotics except colistin (MIC ≤ 2 mcg/mL) or glycylcycline. The data were collected by case record forms.

### 3.5. Outcomes

The primary outcome was the mortality rate of patients with XDRAB pneumonia after treatment with colistin plus sulbactam at dosages of either 9 g/day or 12 g/day at 28 days.

Secondary outcomes were the mortality rates at 7 and 14 days, length of hospital stay, ICU stay, ventilator days (we only included survived patients), microbiological clearance (defined as negative sputum culture on day 7 after treatment), and factors affecting the mortality rate.

The safety outcomes were side effects of colistin and sulbactam such as acute kidney injury according to Kidney Disease Improving Global Outcomes (KDIGO) 2012 guidelines [42], rash, nausea and vomiting, and diarrhea.

### 3.6. Statistical Analysis

The primary null hypothesis was that the dose of 9 g/day of sulbactam would present the same mortality rate as the dose of 12 g/day. Data from previous studies [10,43] found that the mortality rate of XDRAB pneumonia patients treated with colistin combined with a sulbactam dosage of 9 g/day was 73%. This trial was designed as a superiority trial with 80% power to detect a 20-percentage-point absolute difference in mortality rates (e.g., 73% vs. 53%) between the two study groups, with a one-sided alpha level of 0.025. One hundred and thirty-eight participants were required (69 per group). Two planned interim analyses for safety and efficacy were performed either at the end of each year or once 1/3 (46) and 2/3 (92) of the expected number of participants were discharged from the hospital, whichever came first. Group-sequential stopping boundaries were defined with the use of the Lan–DeMets method with an O’Brien–Fleming-type spending function. We report the second planned interim analysis at the end of the second year, including 88 patients. After the interim analysis, we continued to collect data until the desired number of participants was reached.

Kaplan–Meier survival analysis was used to compare the mortality rate between treatment groups by stratified log-rank statistic. The measurement data used the Pearson’s χ^2^ test or the Fisher’s exact test for category outcomes and the Student’s *t* test or the Mann–Whitney U test for continuous outcomes for comparisons across treatment groups. All statistical analyses were performed with STATA software (version 16). These analyses were carried out by an independent statistician using independent data.

## 4. Results

At the end of the second year of the study, we had enrolled 140 patients with XDRAB pneumonia, 88 of whom met the eligible criteria. Forty-three patients were administered sulbactam 9 g/day, and 45 were administered sulbactam 12 g/day (Figure 1).

The baseline characteristics of the patients are shown in the Table 1. The majority of the participants were men, with a mean age of 71.6 years (SD 15.3) and a mean BMI of 22.12 kg/m^2^ (SD 3.83), and most patients were critically ill (defined by ICU admission) (60.23%). The most common underlying diseases were hypertension (75%), followed by diabetes mellitus (46.6%) and dyslipidemia (46.6%). The majority of patients were diagnosed with VAP (77.27%). The disease severity of pneumonia, determined by APACHEII score and SOFA, indicated the mean scores of 25.58 and 8.74. The most commonly administered empirical treatment was colistin in combination with carbapenems, either imipenem or meropenem (87.5%), and only 4.55% of the patients received colistin in combination with sulbactam, two patients at a dose of 9 g/day, and two patients at 12 g/day. (Table 1).

Regarding the MIC (minimum inhibitory concentration) for sulbactam, the mean MIC50 was 32 mcg/mL, and the MIC90 was ≥256 mcg/mL, as shown in Figure 2.

The overall 28-day mortality rate was 52.27%. Trends toward a lower mortality rate were found in the 12 g/day group: at 7 days the mortality rate was 11.1% vs. 23.30% (*p* = 0.17), at 14 days it was 33.3% vs. 41.9% (*p* = 0.27), and at 28 days it was 46.7% vs. 58.1% (*p* = 0.26). The data are shown in Figure 3 and Table 2.

A total of 45 patients in the primary analysis population who were randomized to receive sulbactam at 12 g/day did not meet the primary endpoint of mortality rate compared with 43 patients in the 9 g/day group (risk difference at 7 day, −13.44%; two-sided 95% CI, −30.42 to 3.53; *p* = 0.113, at 14 day, −12.24%; two-sided 95% CI −33.7 to 9.22; *p* = 0.233, at 28 day −12.89%; two-sided 95% CI 34.89 to 9.11; *p* = 0.205) (Figure 4).

The subgroup analysis of 28-day mortality rate related to the sulbactam MIC in patients who received sulbactam at 12 g/day did not show statistically significant differences between the two groups. Better trends in mortality rate were found for subgroups with sulbactam MIC > 48 mcg/mL (42.9% vs. 61.5%, *p* = 0.33) and for patients with sulbactam MIC > 96 mcg/mL (31.5% vs. 63.6%, *p* = 0.23). No statistically significant differences were found between the two subgroups when analyzing the 28-day mortality rate related to septic shock, bacteremia, and DIC. The data are shown in Figure 5 and Figure 6.

The length of hospital stay and the number of ventilator days and ICU days were not statistically different between the two groups (Table 3). Microbiological cure at days 7 defined as negative sputum culture was higher in the 12 g/day group (90.5 vs. 58.1%, *p* = 0.02). The subgroup analysis of patients with sulbactam MIC ≥ 48 mcg/mL showed a higher microbiological cure at day 7 in the 12 g/day group than in the 9 g/day group (93.3% vs. 58.3%, *p* = 0.03), but no statistically significant difference was found for patients with sulbactam MIC < 48 mcg/mL (86.4% vs. 70%, *p* = 0.2). Adverse events after treatment were not significantly different. (Table 4).

In univariate analysis, factors associated with the 28-day mortality rate were asthma (HR 3.59; 95% CI, 1.09–11.81; *p* = 0.04), cirrhosis (HR 4.39; 95% CI, 1.53–12.56; *p* = 0.01), APACHE score ≥ 28 (HR 3.57; 95% CI, 1.99–6.40; *p* =< 0.01), SOFA score ≥ 9 (HR 1.93; 95% CI, 1.05–3.55; *p* = 0.03), and DIC (HR 1.84; 95% CI, 1.03–3.29; *p* = 0.04). In multivariate analysis, factors associated with 28-day mortality were asthma (adjusted HR 4.69; 95% CI, 1.22–18.09; *p* = 0.03), cirrhosis (adjusted HR 3.8; 95% CI, 1.06–13.57; *p* = 0.04), APACHEII score ≥ 28 (adjusted HR 2.94; 95% CI, 1.51–5.72; *p* < 0.01), sulbactam dosage of 9 g/day (adjusted HR 2.02; 95% CI, 1.10–3.71; *p* = 0.02), as shown in Table 5.

## 5. Discussion

In this randomized controlled trial, the administration of colistin in combination with a high dose of sulbactam at either 9 or 12 g/day, did not result in significant differences in mortality rate, length of hospital stay, and number of ventilator days and ICU days. However, the sulbactam dosage of 12 g/day showed a trend toward lowering the mortality rate in XDRAB pneumonia patients at 7, 14, and 28 days.

XDRAB pneumonia remains a significant problem due to its high mortality rate, despite many studies trying drug or dose adjustments, and XDRAB is responsible for the current increase in drug resistance [5,6]. Previous studies using sulbactam at the dosages of 3 and 9 g/day also found a 28-day mortality rate of more than 70% [10,44].

In this study, a dosage of sulbactam of at least 9 g/day with prolonged infusion for 4 h, was used in accordance with the European Society of Intensive Care Medicine recommendations [33]. However, mortality rates were still comparable to those of previous studies which did not use prolonged infusion, i.e., 70% with 3 g/day [44] and 51.9% with 9 g/day [10]. The overall mortality rate was lower in this study when administering the higher dose of sulbactam in spite of the approximate APACHE score. However, this study found a lower mortality rate at 12 g/day, corresponding to 46.7%, compared to that at 9 g/day, corresponding to 58.1%.

This may be due to the following factors: most of the participants were elder (the average age was 71.6 years) than in previous studies [10,44,45], and patients were critically ill and had high APACHEII or SOFA scores. Comparing the mortality rates at 7, 14, 28 days, there was a tendency to a lower value in the 12 g/day group, but the higher number of septic shock participants in this group may have led to a higher mortality rate. In addition, only 88 patients participated (rather than 138, as calculated) in this study, and this number may have been insufficient to determine a statistically significant difference in mortality rate.

In this study, we found a tendency to a lower mortality rate when administering a higher dose of sulbactam. This was also found in other studies that determined a lower mortality rate with a higher dose of sulbactam. Our results of mortality rates are comparable to those of previous studies, despite slightly differences in the participants. The 28-day mortality rate of colistin in combination with sulbactam at a dosage of 12 g/day (46.7%) was lower than that found in previous studies which used lower doses of sulbactam, i.e., 3 g/day (70%) [44], 6 g/day (51.1%) [45], and 9 g/day (51.9%) [10]. Therefore, using a higher dose of sulbactam may reduce the mortality rate in XDRAB pneumonia patients.

Although all XDRAB isolates in this study were susceptible to colistin, sulbactam may play a role especially in subpopulations showing heteroresistance to colistin. Previous studies reported that some XDRAB isolates with colistin MIC ≤ 2 mcg/mL may contain subpopulations of heteroresistance to colistin which can be difficult to detect by standard AST [46,47].

The microbiological cure was significantly higher in the 12 g/day group (90.5%) compared with the 9 g/day group (58.1%), consistent with previous studies that found a lower cure rate at the lower sulbactam dosages of 9 g/day (85.7%) [10], 6 g/day (84%) [29], and 3 g/day (60%) [44]. However, the cure rate for the group receiving 9 g/day of sulbactam was lower than in the previous studies. This may be due to a higher MIC of sulbactam for XDRAB than in previous studies. The MIC90 was only 16 μg/mL in 1988–1999, rising to 192 μg/mL in 2013–2016 [34]; in this study, the MIC90 was ≥256 μg/mL. Further studies should evaluate the outcomes of the optimal sulbactam dosage in regard to the MIC of individual patients.

Acute kidney injury was still a common complication of colistin treatment, which was 30.68%, comparable to the study by Koomanachai P, Thamlikitkul V, et al. [43]. From this study, the dose of sulbactam at 9 g/day compared to 12 g/day, showed no statistically significant differences in renal outcome. Consequently, sulbactam at a dose of 12 g/day was shown to be safe and did not increase the incidence of adverse events.

A high APACHEII score was also found to still be a reliable predictor of higher mortality. Underlying diseases, asthma, and cirrhosis, increase the risk of death in XDRAB pneumonia patients. A sulbactam dosage of 9 g/day was found to be a risk factor for high mortality in XDRAB pneumonia patients with a high MIC for sulbactam. Further study of MIC and optimum dosage of sulbactam in XDRAB pneumonia patients with high sulbactam MIC is needed in the future.

Despite this study being the first randomized control trial of colistin in combination with sulbactam at 12 g/day to report patients’ mortality rate, it has certain limitations. First, due to the limited study time period, the patient population did not reach the target number. Insufficient participants may have resulted in the lack of difference in clinical outcome between the two groups. Second, the upper limit of MIC was 256; therefore, the value of MIC90 could not be determined clearly. This may have affected the determination of the appropriate dose of sulbactam. Third, this study considered a maximum dose of sulbactam of 12 g/day, but the MIC of sulbactam is currently increasing. Further studies should be conducted to determine if a higher sulbactam dosage is contributing to the higher MIC. The last limitation is that a formulation of sulbactam without ampicillin or cefoperazone may not be available in some countries. Therefore, high doses of ampicillin/sulbactam or cefoprazone/sulbactam should be examined to determine their possible side effects.

## 6. Conclusions

In summary, XDRAB pneumonia patients at Phramongkutklao Hospital have a high mortality rate. Treatment with colistin combined with sulbactam at 12 g/day was not superior to combination treatment with sulbactam at 9 g/day, in regard to mortality at any timepoint in XDRAB pneumonia patients with high sulbactam MIC. The value of MIC90 in this study could not be determined clearly because the upper limit of MIC was 256. Further studies are needed to determine the MIC of sulbactam and the optimum dosage of sulbactam.

## Figures and Tables

**Figure 1 antibiotics-11-01112-f001:**
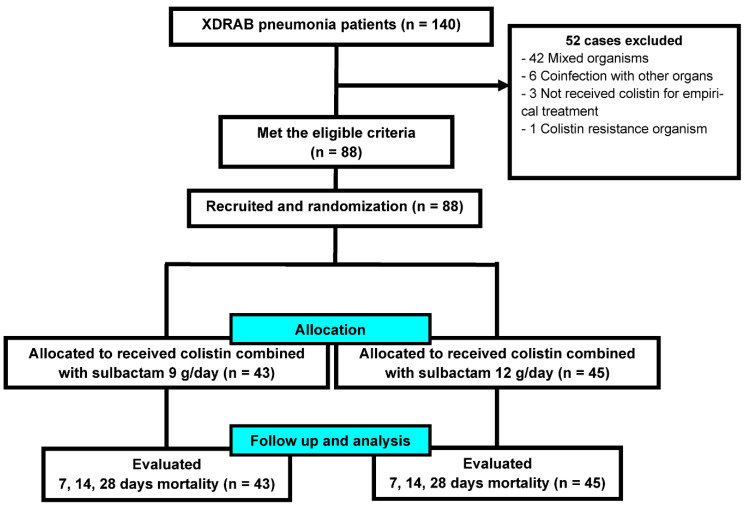
Enrollment and follow-up of the participants.

**Figure 2 antibiotics-11-01112-f002:**
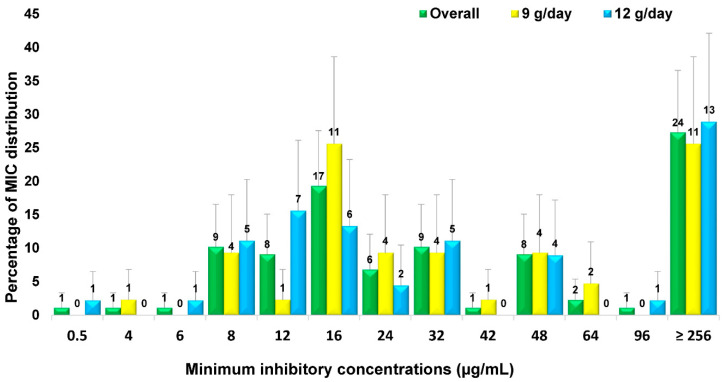
Distribution of sulbactam MIC for XDRAB. The value labels indicate the number of isolates. Error bars indicate the 95% confidence interval.

**Figure 3 antibiotics-11-01112-f003:**
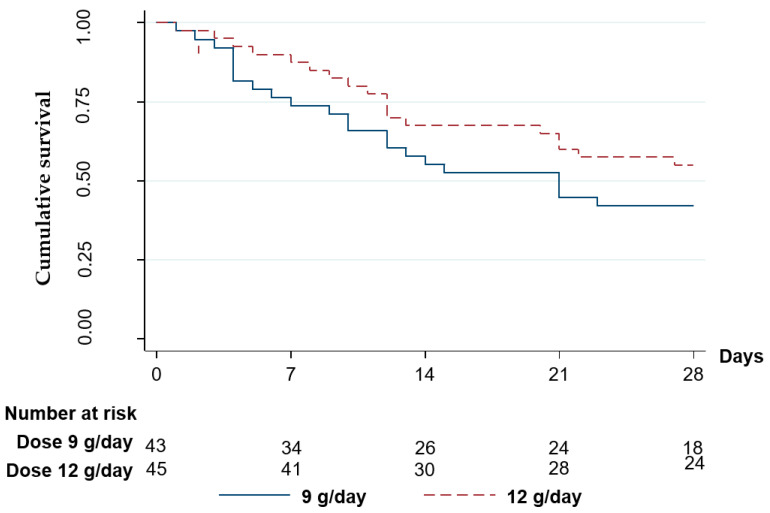
Kaplan–Meier survival curves for 28-day mortality of XDRAB pneumonia patients in relation to the administered dose of sulbactam.

**Figure 4 antibiotics-11-01112-f004:**
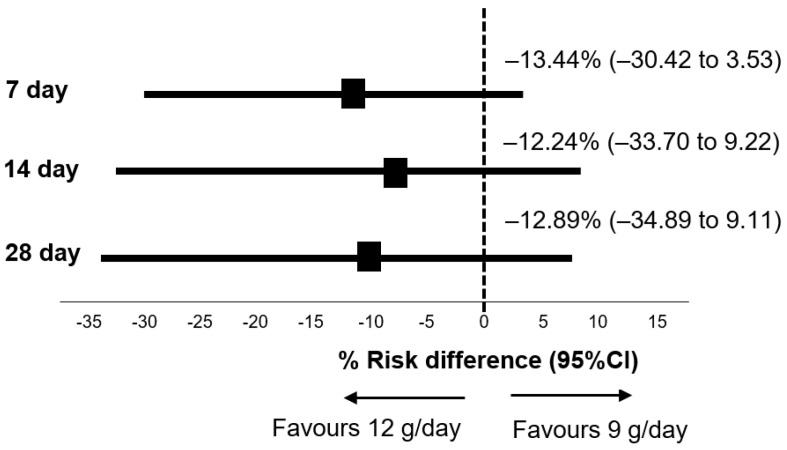
Risk difference for mortality at day 7, 14, 28 in patients treated with 12 g/day vs. 9 g/day of sulbactam. Abbreviation: CI, confidence interval.

**Figure 5 antibiotics-11-01112-f005:**
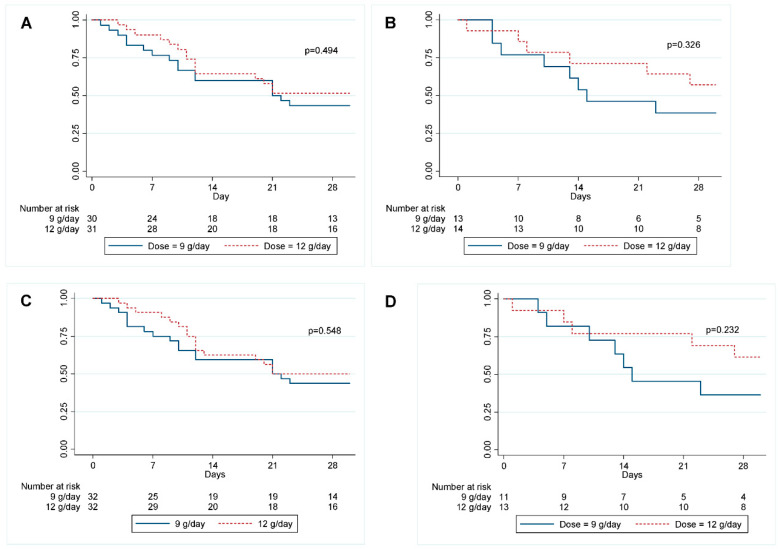
Kaplan–Meier survival curves for 28-day mortality of XDRAB pneumonia patients in relation to dose with subgroup analysis of MIC of sulbactam. (**A**) MIC of sulbactam ≤ 48 mcg/mL, (**B**) MIC of sulbactam > 48 mcg/mL (**C**) MIC of sulbactam ≤ 96 mcg/mL, (**D**) MIC of sulbactam > 96 mcg/mL.

**Figure 6 antibiotics-11-01112-f006:**
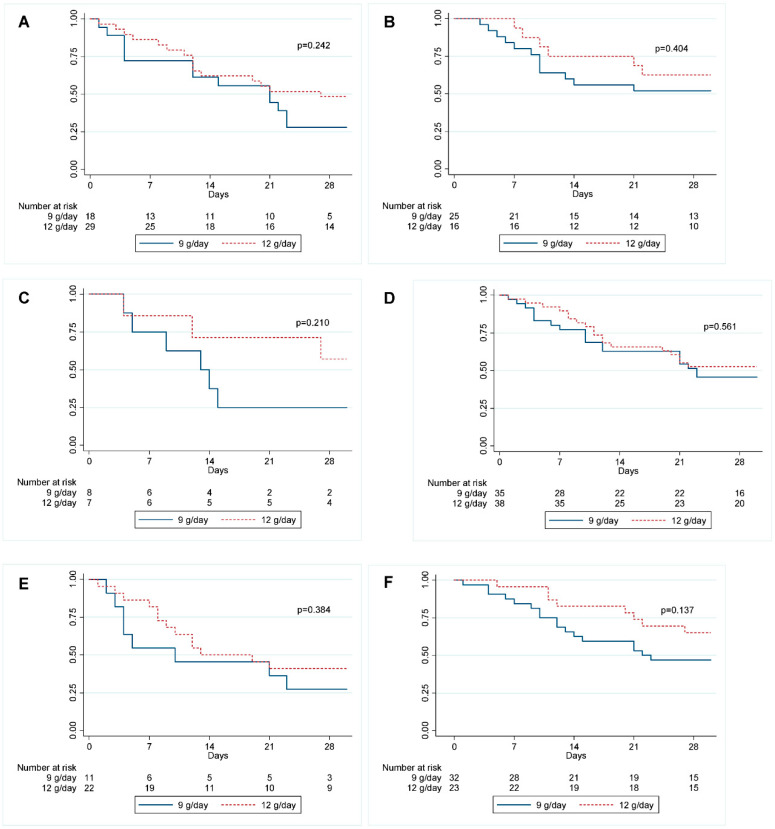
Kaplan-Meier survival curves for 28-day mortality in XDRAB pneumonia patients in relation to sulbactam dose, with subgroup analysis of complications from pneumonia. (**A**) Pneumonia with septic shock, (**B**) pneumonia without septic shock, (**C**) pneumonia with bacteremia, (**D**) pneumonia without bacteremia, (**E**) pneumonia with DIC, (**F**) pneumonia without DIC.

**Table 1 antibiotics-11-01112-t001:** Baseline characteristics.

	Colistin/Sulbactam 9 g/Day (n = 43)	Colistin/Sulbactam 12 g/Day (n = 45)
**Male sex**, n (%)	30/43 (69.8%)	35/45 (77.8%)
**Age**, years; mean ± SD	75.35 ± 12.85	67.84 ± 17.74
**BMI**, kg/m^2^; mean ± SD	22.7 ± 3.92	21.53 ± 3.74
**ICU Ward**, n (%)	22/43 (51.2%)	31/45 (68.9%)
**Underlying disease**, n (%)		
Diabetes mellitus	23/43 (53.5%)	18/45 (40.0%)
Hypertension	35/43 (81.4%)	31/45 (68.9%)
Dyslipidemia	21/43 (48.8%)	20/45 (44.4%)
COPD	4/43 (9.3%)	2/45 (4.4%)
Asthma	1/43 (2.3%)	2/45 (4.4%)
Chronic kidney disease		
Stage 3	5/43 (11.6%)	6/45 (13.3%)
Stage 4 and 5	4/43 (9.3%)	1/45 (2.2%)
ESRD	7/43 (16.3%)	7/45 (15.6%)
Cirrhosis	2/43 (4.7%)	2/45 (4.4%)
Gout	1/43 (2.3%)	2/45 (4.4%)
Ischemic heart disease	7/43 (16.3%)	8/45 (17.8%)
Stroke	9/43 (20.9%)	5/45 (11.1%)
Malignancy	6/43 (14%)	11/45 (24.4%)
**Diagnosis**, n (%)		
HAP	12/43 (27.9%)	8/45 (17.8%)
VAP	31/43 (72.1%)	37/45 (82.2%)
**APACHE II score**, n (%)		
≤19	7/43 (16.3%)	7/45 (15.6%)
20–24	12/43 (27.9%)	15/45 (33.3%)
25–29	16/43 (37.2%)	10/45 (22.2%)
≥30	8/43 (18.6%)	13/45 (28.9%)
**SOFA score**, n (%)		
≤6	19/43 (44.2%)	10/45 (22.2%)
7–9	11/43 (25.6%)	11/45 (24.4%)
10–12	5/43 (11.6%)	14/45 (31.1%)
≥13	8/43 (18.6%)	10/45 (22.2%)
**Complication from pneumonia**, n (%)		
Septic shock	18/43 (41.9%)	29/45 (64.4%)
Bacteremia	8/43 (18.6%)	7/45 (15.6%)
DIC	11/43 (25.6%)	22/45 (48.9%)
**Empirical antibiotics**, n (%)		
Carbapenems	35/43 (81.4%)	42/45 (93.3%)
Sulbactam	2/43 (4.7%)	2/45 (4.4%)
Others	6/43 (14.0%)	1/45 (2.2%)

**Abbreviations**: BMI, body mass index; ICU, intensive care unit; COPD, chronic obstructive pulmonary disease; ESRD, end stage renal disease; HAP, hospital-acquired pneumonia; VAP, ventilator-associated pneumonia; APACHE, acute physiology and chronic health evaluation; SOFA, sequential organ failure assessment; DIC, disseminated intravascular coagulation.

**Table 2 antibiotics-11-01112-t002:** Seven, 14, 28 days survival of XDRAB pneumonia patients in relation to the administered dose of sulbactam.

Days	9 g/day	12 g/day	*p*-Value
	Survivor Rate % (95% CI)	Survivor Rate % (95% CI)	
**7**	76.7 (61.1–86.8)	88.9 (75.3–95.2)	0.17
14	58.1 (42.1–71.2)	66.7 (50.9–78.4)	0.27
28	41.9 (27.1–55.9)	53.3 (37.9–66.6)	0.26

*p*-value by Log rank test.

**Table 3 antibiotics-11-01112-t003:** Length of hospital stay, number of ventilator days and ICU days, and microbiological cure.

	Colistin/Sulbactam 9 g/Day	Colistin/Sulbactam 12 g/Day	*p*-Value
**Length of stay, days**; median (95% CI)	31 (19, 43)	36 (7, 71)	0.08
**Ventilator days, days**; median (95% CI)			
HAP group	3 (1, 8)	13 (1, 29)	0.13
VAP group	19 (4, 34)	23 (19, 27)	0.46
**ICU days**, days; median (95% CI)	14 (12, 16)	17 (6, 28)	0.33
**Microbiological cure** at day 7, n (%)	25/43 (58.1%)	38/42 (90.5%)	0.02

**Abbreviations**: ICU, intensive care unit; HAP, hospital-acquired pneumonia; VAP, ventilator-associated pneumonia.

**Table 4 antibiotics-11-01112-t004:** Adverse events.

Adverse Events, n (%)	Colistin/Sulbactam 9 g/Day (n = 43)	Colistin/Sulbactam 12 g/Day (n = 45)	*p*-Value
Any adverse events	17/43 (39.5%)	17/45 (37.8%)	0.87
AKI	12/43 (27.9%)	15/45 (33.3%)	0.58
Diarrhea	5/43 (11.6%)	2/45 (4.4%)	0.21

**Abbreviations**: AKI, acute kidney injury.

**Table 5 antibiotics-11-01112-t005:** Univariate and Multivariate Analyses of Risk Factors Associated with 28-day Mortality Using the Cox Regression Model.

Variables	Crude Analysis		Adjusted Analysis	
	HR (95%CI)	*p*-Value	HR (95%CI)	*p*-Value
**Underlying disease**				
Asthma	3.59 (1.09, 11.81)	0.04	4.69 (1.22, 18.09)	0.03
Cirrhosis	4.39 (1.53, 12.56)	0.01	3.8 (1.06, 13.57)	0.04
**APACHE score ≥ 28**	3.57 (1.99, 6.40)	<0.01	2.94 (1.51, 5.72)	<0.01
**SOFA score ≥ 9**	1.93 (1.05, 3.55)	0.03	1.71 (0.58, 5.01)	0.33
**Dosage of** **Sulbactam**				
9 g/day	1.39 (0.78,2.47)	0.27	2.02 (1.1, 3.71)	0.02
12 g/day	Reference	1	Reference	1
**Complication from pneumonia**				
DIC	1.84 (1.03, 3.29)	0.04	1.53 (0.79, 2.96)	0.21

**Abbreviations**: APACHE, acute physiology and chronic health evaluation; SOFA, sequential organ failure assessment; DIC, disseminated intravascular coagulation.

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
