# Peer review of "A Randomized Controlled Trial of Colistin Combined with Sulbactam: 9 g per Day versus 12 g per Day in the Treatment of Extensively Drug-Resistant Acinetobacter baumannii Pneumonia: An Interim Analysis"

_antibiotics, 2022, doi:10.3390/antibiotics11081112_

Round 1
Reviewer 1 Report
This manuscript by Ungthammakhun et al describes a controlled trial in its second year which seeks to determine the effective dosage of sulbactam for use in combination with colistin against XDR Acinetobacter. This is a significant healthcare burden and so any treatment options are important to study. While only partway through the trial, there is a trend towards the highest dose of 12g/day as the most effective dosage. The manuscript is well written and includes all relevant data. Only a couple of minor points below, but would otherwise recommend acceptance.
Minor points:
-How is XDR Acinetobacter defined in this study? Were the isolates identified as Acinetobacter by MALDI-TOF or some other method? How was antimicrobial susceptibility testing (AST) done? And what are the specific resistance criteria to consider an isolate XDR?
-If AST data is available it would be interesting to see which antibiotics the isolates were resistant/susceptible to. Were there any other antibiotics besides colistin to which these isolates were susceptible?
-Acinetobacter is known to have high rates of heteroresistance to colistin (Sherman EX et al 2019), so this could also be discussed in the text as a possible confounder in some patients, as heteroresistance can be difficult to detect by standard AST
Author Response
Response to Reviewer 1
Thank you very much for reviewing of our articles. We have answered your points below.
- How is XDR Acinetobacter defined in this study? Were the isolates identified as Acinetobacter by MALDI-TOF or some other method? How was antimicrobial susceptibility testing (AST) done? And what are the specific resistance criteria to consider an isolate XDR?
Response:
Species were identified by MALDI-TOF (Bruker Daltonics). The method for measuring the MIC values was according to the Clinical & Laboratory Standards Institute (CLSI) guidelines for sulbactam and colistin of XDRAB. Sulbactam MIC were obtained by Epsilometer test (E-test, Liofilchem). The automated broth microdilution test (Sensititre, Thermo Fisher) was use for MIC breakpoint of other antibiotics including colistin. XDRAB was defined when an isolate resists to all antibiotics except colistin (MIC ≤2 µg/mL) or glycylcycline. The data were collected by case record forms.
We have included these details in the manuscript as suggested.
- If AST data is available it would be interesting to see which antibiotics the isolates were resistant/susceptible to. Were there any other antibiotics besides colistin to which these isolates were susceptible?
Response: Apart from colistin, all isolates of XDRAB in this study were susceptible to tigecycline and resistant to others.
- Acinetobacter is known to have high rates of heteroresistance to colistin (Sherman EX et al 2019), so this could also be discussed in the text as a possible confounder in some patients, as heteroresistance can be difficult to detect by standard AST.
Response: Thank you very much for your reflection of the important point that may affect the results of this study. These points were discussed further in the manuscript (page 12 ,line 356-360).

Reviewer 2 Report
Although the study is underpowered to detect the difference between the two groups, the trend is still intertesting and informative. I would like to ask/add some issues as below:
1. Please explain for the exclusion criteria of the study for colistin use before enrollment to be 5 days. (Why not lesser?)
2. It would be interesting to see the difference between the two groups in the subjects whose XDRABs were sensitive to sulbactam for the mortality and the microbiologic cure rate at 7 days.
3. Look forward to see the results of the study with full enrollment in the future.
Author Response
Response to Reviewer 2
Thank you very much for reviewing of our articles. We have answered your points below
- Please explain for the exclusion criteria of the study for colistin use before enrollment to be 5 days. (Why not lesser?)
Response: Species identification and susceptibility testing in the study site take approximately 3-5 days. Therefore, participants who were given colistin before the enrolment more than 5 days were excluded to minimize the possible confounding effect of prior use of colistin monotherapy. All participants in the study received the same colistin at empirical treatment.
- It would be interesting to see the difference between the two groups in the subjects whose XDRABs were sensitive to sulbactam for the mortality and the microbiologic cure rate at 7 days.
Response: Thank you for another important and interesting point. It is very interesting that this study showed a more microbiological cure rate of higher dose of sulbactam at 7 day especially in MIC<48 subgroup. We have added this data in the manuscript.
- Look forward to see the results of the study with full enrollment in the future.
Response: Thanks for your attention to our articles and I hope this study will be published. We expect to completely enrollment of the participants soon.

Reviewer 3 Report
The present study aims to evaluate a combination therapy of colistin with sulbactam at a dosage of 9g/day v/s 12 g/day for the treatment of XDRAB pneumonia. The attempts made by the authors are noteworthy, however, the current manuscript is not suitable for publication because of the following concerns:
Major concerns:
1. The statistical significance of the study across 7 to 28 days of post colistin and sulbactam treatment is not convincing between the 9 g/day sulbactam and 12 g/day sulbactam group. The sample size for the study is small which might lead to biasness. The authors themselves suggest in the discussion (line 400) that the study population did not reach the targeted number which might have resulted in no difference in the clinical outcome. In this regard, this manuscript will have more scientific significance when the study is completed with the planned number of participants (69 per group).
2. More explanation for the rationale of colistin-Sulbactam combination need to be described. Considering that Sulbactam is a beta-lactamase inhibitor, it will be ideal to combine it with a beta-lactam antibiotic or other bacterial cell-wall synthesis inhibiting antibiotics in addition to colistin. For instance, the reference (22) cited in the introduction clearly suggests that triple combination of meropenem (a beta lactam antibiotic that inhibits bacterial cell wall peptidoglycan biosynthesis), sulbactam and colistin showed synergy against 96.7 percent of MDR A. baumannii while double combination with sulbactam and colistin showed only 53.3 % synergy. Similarly, another study (reference 21) also mentions that rates of synergistic and additive effect rates of colistin plus sulbactam and colistin plus Fosfomycin (a bacterial cell wall peptidoglycan biosynthesis inhibiting antibiotic) were 53.3% and 73.3% of isolates, respectively. These references cited in the introduction in support for the colistin-sulbactam combination applied in this work itself clearly indicate that this combination has lesser synergistic outcomes. Moreover, the group with colistin/sulbactam 12g/day (n=45) was having more patients subject to empirical antibiotic treatment with Carbapenems (42 out of 45 - 93%). This was only 35 out of 43 (81.4%) in the colistin/sulbactam 9 g/day group. The observed significant differences at day 7 microbiological cure rate in colistin/sulbactam 12g/day may be attributed to this aspect.
3. Does the statement in the abstract (line 19) “Treatment with colistin combined with sulbactam at 12 g/day was not superior to 9 g/day” does this implies to all time points investigated (i.e day 7, 14, 28) or to 28 days alone. It will be useful to mention it along with what is referred to. Because, in the discussion it is mentioned that “dosage of sulbactam 12 g/d had trends toward lowering the mortality rate in XDRAB pneumonia patients” whereas in the conclusion it is mentioned that “There was no statistically significant difference in mortality rates between colistin combined with sulbactam at 9 g/day and 12 g/day in the high MIC of sulbactam in XDRAB pneumonia patients.
Minor concerns:
1. The title of the manuscript has a ‘extensively drug-resistant resistant Acinetobacter baumannii’ =>> extensively drug-resistant Acinetobacter
2. Page 1, Lines 24: Ref 2 is cited before Ref 1. It should be checked.
3. The legend for Figure 2 must be more elaborative. For example, the number of replicates done and error bars are lacking.
4. In Figure 3, the labelling of the Y-axis is not indicated.
5. In Figure 4, more details of the tool used to arrive at the chart needs to be described in the legend.
6. Ampicillin/cefoperazone is referred to at the end of the discussion (line 407) without any background info. It will be useful to include additional information with suitable references.
Author Response
Response to Reviewer 3
Thank you very much for reviewing of our articles. We have answered your points below
Major concerns:
- The statistical significance of the study across 7 to 28 days of post colistin and sulbactam treatment is not convincing between the 9 g/day sulbactam and 12 g/day sulbactam group. The sample size for the study is small which might lead to biasness. The authors themselves suggest in the discussion (line 400) that the study population did not reach the targeted number which might have resulted in no difference in the clinical outcome. In this regard, this manuscript will have more scientific significance when the study is completed with the planned number of participants (69 per group).
Response: Thank you for mentioning in study population. The widespread Covid-19 outbreak in Thailand has made data collection slower than it should. Therefore, this interim analysis was performed at the end of second year. After the interim analysis we have been collecting data with an expectation to complete the enrollment soon.
- More explanation for the rationale of colistin-Sulbactam combination need to be described. Considering that Sulbactam is a beta-lactamase inhibitor, it will be ideal to combine it with a beta-lactam antibiotic or other bacterial cell-wall synthesis inhibiting antibiotics in addition to colistin. For instance, the reference (22) cited in the introduction clearly suggests that triple combination of meropenem (a beta lactam antibiotic that inhibits bacterial cell wall peptidoglycan biosynthesis), sulbactam and colistin showed synergy against 96.7 percent of MDR A. baumannii while double combination with sulbactam and colistin showed only 53.3 % synergy. Similarly, another study (reference 21) also mentions that rates of synergistic and additive effect rates of colistin plus sulbactam and colistin plus Fosfomycin (a bacterial cell wall peptidoglycan biosynthesis inhibiting antibiotic) were 53.3% and 73.3% of isolates, respectively. These references cited in the introduction in support for the colistin-sulbactam combination applied in this work itself clearly indicate that this combination has lesser synergistic outcomes. Moreover, the group with colistin/sulbactam 12g/day (n=45) was having more patients subject to empirical antibiotic treatment with Carbapenems (42 out of 45 - 93%). This was only 35 out of 43 (81.4%) in the colistin/sulbactam 9 g/day group. The observed significant differences at day 7 microbiological cure rate in colistin/sulbactam 12g/day may be attributed to this aspect.
Response: Although sulbactam is a beta-lactamase inhibitor (BI), it has a unique activity comparable to beta-lactam (BL) against A. baumannii observed through in vitro studies, animal models, and clinical outcomes data. The BL activity is through the binding of penicillin binding protein 1 (PBP1) and PBP3. The current 2022 IDSA guidance on the treatment of carbapenem-resistant A. baumannii suggested in favor of the combination therapy with at least 2 agents of in vitro activity such as polymyxin, high dose of ampicillin-sulbactam, or tigecycline whereas against the use of polymyxin in combination with meropenem. This recommendation of ampicillin-sulbactam instead of sulbactam was because sulbactam is not available in the US. As suggested, we have included studies of sulbactam in combination with colistin without cell-wall active agents in the introduction (page 2).
According to pharmacokinetics and pharmacodynamics of BI, the synergistic or additive effect of BI in addition to BL occurs only during the time where serum concentrations of both drugs are within therapeutic range. Because carbapenems were discontinued before the administration of sulbactam, the effect of sulbactam to prior given carbapenems, if there was, should be minimal.
- Does the statement in the abstract (line 19) “Treatment with colistin combined with sulbactam at 12 g/day was not superior to 9 g/day” does this implies to all time points investigated (i.e day 7, 14, 28) or to 28 days alone. It will be useful to mention it along with what is referred to. Because, in the discussion it is mentioned that “dosage of sulbactam 12 g/d had trends toward lowering the mortality rate in XDRAB pneumonia patients” whereas in the conclusion it is mentioned that “There was no statistically significant difference in mortality rates between colistin combined with sulbactam at 9 g/day and 12 g/day in the high MIC of sulbactam in XDRAB pneumonia patients.
Response: We have included the timepoint for clarification as suggested.
Minor concerns:
- The title of the manuscript has a ‘extensively drug-resistant resistant Acinetobacter baumannii’ =>> extensively drug-resistant Acinetobacter
Response: Thank you for the suggestion. We have corrected the error.
- Page 1, Lines 24: Ref 2 is cited before Ref 1. It should be checked.
Response: We have revised the text and reference (page 1, line 24).
- The legend for Figure 2 must be more elaborative. For example, the number of replicates done and error bars are lacking.
Response: We have revised the figure 2 as suggestion.
- In Figure 3, the labelling of the Y-axis is not indicated.
Response: We have revised the Figure 3 by labelling of the Y-axis (page 7).
- In Figure 4, more details of the tool used to arrive at the chart needs to be described in the legend.
Response: We have revised the Figure 4 : Risk difference of mortality rate at day 7, 14, 28 in patients treated with 12 g/day VS 9 g/day of sulbactam. (page 8)
- Ampicillin/cefoperazone is referred to at the end of the discussion (line 407) without any background info. It will be useful to include additional information with suitable references.
Response: In many countries, sulbactam is only available in combination regimen such as ampicillin/sulbactam or cefoperazone/sulbactam. We have clarified further in the manuscript as suggested.

Round 2
Reviewer 3 Report
After considering the author's response, this reviewer recommends this manuscript to be published.
In Introduction, para 5: “The study of sulbactam 9 and 12 g/day found no significant complications compared with colistin monotherapy.(11, 14)”, have actually used ampicillin-sulbactam combination and not sulbactam alone. Please modify the text accordingly.
Author Response
Thank you very much for reviewing of our articles again. We have modified the details in the manuscript as suggested
